# Impacts of Neonicotinoids on Molluscs: What We Know and What We Need to Know

**DOI:** 10.3390/toxics9020021

**Published:** 2021-01-22

**Authors:** Endurance E Ewere, Amanda Reichelt-Brushett, Kirsten Benkendorff

**Affiliations:** 1Marine Ecology Research Centre, School of Environment, Science and Engineering, Southern Cross University, P.O. Box 157, Lismore, NSW 2480, Australia; endurance.ewere@uniben.edu (E.E.E.); amanda.reichelt-brushett@scu.edu.au (A.R.-B.); 2Department of Animal and Environmental Biology, Faculty of Life Sciences, University of Benin, PMB 1154 Benin City, Nigeria; 3National Marine Science Centre, School of Environment, Science and Engineering, Southern Cross University, 2 Bay Drive, Coffs Harbour, NSW 2450, Australia

**Keywords:** non-target species, toxicity, biomarker, pesticide, bivalve, gastropod, cephalopod, environmental concentration

## Abstract

The broad utilisation of neonicotinoids in agriculture has led to the unplanned contamination of adjacent terrestrial and aquatic systems around the world. Environmental monitoring regularly detects neonicotinoids at concentrations that may cause negative impacts on molluscs. The toxicity of neonicotinoids to some non-target invertebrates has been established; however, information on mollusc species is limited. Molluscs are likely to be exposed to various concentrations of neonicotinoids in the soil, food and water, which could increase their vulnerability to other sources of mortality and cause accidental exposure of other organisms higher in the food chain. This review examines the impacts of various concentrations of neonicotinoids on molluscs, including behavioural, physiological and biochemical responses. The review also identifies knowledge gaps and provides recommendations for future studies, to ensure a more comprehensive understanding of impacts from neonicotinoid exposure to molluscs.

## 1. Introduction

Rapid population growth, in combination with concerns regarding food security for humans, has led to an increase in the production and utilisation of pesticides [1,2] to control damaging insects in agriculture and improve food production [3]. While this has been successful based on primary intentions, the increasing use of pesticides has caused several deleterious effects on non-target organisms in the environment [4], threatening their survival and existence, especially when interacting with other natural environmental stressors. For this reason, in the future, food production will either require alternatives to pesticides, the use of pesticides that are extremely selective at low doses, and effective management of all the stages of agricultural production to ensure that non-target species in the environment are not directly or indirectly affected by pesticides [4,5].

Several insecticides are used to control crop-damaging insects in agricultural programs around the world, helping to minimise crop loss from pests and diseases [6]. Insecticide contamination of terrestrial [7,8,9] and aquatic [10,11] ecosystems have been extensively documented [12,13]. This has led to regulatory controls, including restricted use, revoking approval and bans of certain pesticides in some countries, whilst the same pesticides are actively used in many other countries [14,15].

Neonicotinoids are a group of neuroactive insecticides that were first registered for agricultural use in the 1990s and just two decades later they were among the most widely used insecticides in the world [16], accounting for one-third of the global market for insecticides [17]. However, at the time neonicotinoids were introduced into the pesticide market, the risk assessment protocols were not sufficient to detect some of the environmental risks associated with these chemicals [17]. Following a moratorium on the use of three neonicotinoids in the European Union in 2013 [18], they have been banned for use in open-field crops in the EU since 2018 [19] and Canada since 2019 [20]. In 2019, the Fijian government voted for a ban on the importation and use of a neonicotinoid (imidacloprid) which took effect from January 2020 [21]. The restricted use was largely due to well documented negative impacts on pollinating insects. However, residues of these neonicotinoid compounds are increasingly being detected in receiving environments within agricultural catchments [11,22,23], triggering further monitoring and the assessment of risks to aquatic invertebrates [20]. The temporal and spatial scale of risks associated with pesticide use needs to be considered across both terrestrial and aquatic environments, with high efficacy and systemicity, long persistence and high mobility identified as the key risks associated with neonicotinoid exposure [17].

The most common pathways for offsite contamination by neonicotinoids are leaching, soil and water run-off [24], foliar deposition [25] and mechanical methods, for instance, spray-drift. Bonmatin et al. [26] conducted a survey of the levels of neonicotinoids residues along a gradient from treated crop fields to adjacent fields and then into aquatic ecosystems. The results showed varying concentrations of neonicotinoids residues in all the environmental samples, with higher concentrations detected closer to the treated zones. Similarly, a concentration of up to 320 µg/L was reported in natural water in the Netherlands [27]. Furthermore, Morrissey et al. [28] reviewed the reported concentrations of neonicotinoids in aquatic systems in nine countries and found these to exceed the interim short-term (0.2 µg/L) and long-term (0.035 µg/L) water quality threshold at some locations [28]. These studies indicate that non-target invertebrates (e.g., molluscs) could be exposed to various concentrations of neonicotinoid insecticides, with the likelihood of detrimental impacts [4,28].

Due to the habitat and method of feeding, molluscs could be exposed to various concentrations of neonicotinoids in the environment, with potential negative impacts on the exposed species and the ecosystem structure in general. In fact, neonicotinoids cause stress to terrestrial molluscs [29] and build up in the tissues at concentrations that could cause mortality in mollusc-eating arthropods [29]. Exposure of filter-feeding bivalves to neonicotinoids also led to a build-up of residues in the tissues [30,31] at concentrations that could impact consumers like crabs, crayfish [32,33] and humans [34]. Exposure of Sydney rock oysters to various concentrations (0.01–2 mg/L) caused a wide range of behavioural, biochemical and physiological impacts [30,31,35]. This means that contamination of aquatic systems by neonicotinoids could have significant implications for commercial molluscs (e.g., oysters, scallops, mussels and clams), threatening productivity.

Shellfish reefs provide important services globally with an economic value of more than US$ 40 billion annually [36]. However, over 80% of shellfish reefs have been functionally lost on a global scale [37]. Because of this, billions of dollars are currently being spent on shellfish reef restoration to improve ecosystem services. The full causes for the loss of previously expansive shellfish reefs are unclear; however, water quality is often regarded as an important factor [38,39]. Therefore, this review focuses on the impacts of neonicotinoids exposure on mollusc species. The aim is to highlight what is already known about the response of molluscs to neonicotinoids and to identify knowledge gaps, and on how they can be addressed in future studies.

## 2. Neonicotinoids

### 2.1. Chemical Properties, Registration, Use and Efficacy

A defining characteristic of commercially available neonicotinoids is the presence of at least one sp^3^ nitrogen, either as part of a heterocyclic ring or an acyclic moiety (Figure 1). This sp^3^ nitrogen, in association with a conjugated electron-withdrawing group, led to the definition for “neonicotinoid” [40], and remains central to the neonicotinoid pharmacophore [41].

Neonicotinoids are found in many commercially available insecticide products, for instance, Advocate, Confidor and Admire (imidacloprid), Actra, Endigo and Durivo (thiamethoxam) and Assil, Intruder and Saurus (acetamiprid) (Table 1) which are broadly used to control agricultural and domestic insects [42,43,44]. Owing to their presumed low toxicity to mammals compared to other pesticides and high potency against target insects [45], these insecticides are registered in more than 120 nations [28,41]. Neonicotinoids are one of the most effective pesticides available for the control of sucking insect pests, for example, whiteflies, aphids, thrips, a number micro Lepidoptera, leaf- and plant-hoppers and some beetles [41,46]. They are used for a variety of applications such as veterinary medicine as ectoparasiticides, urban landscaping [41] and as agents for crop protection in many agricultural systems [47]. Imidacloprid and thiamethoxam, for instance, can be applied by numerous techniques including root drench to the soil, foliar sprays over ground plants, or as trunk injection to trees; however, the majority of all neonicotinoids are conveyed as seed/soil treatment in agricultural systems [41,47]. The application rate of neonicotinoids depends on several factors, including the crop type, method of application and the specific neonicotinoid and the country of used [48].

Neonicotinoids bind to the nicotinic acetylcholine receptors (nAChRs) in the central nervous system (CNS) of animals. nAChRs are among the family of ligand-gated ion channels in charge of quick excitatory cholinergic neurotransmission in CNS [41,49]. Neonicotinoids are thought to bind to the nAChRs of insects with much higher efficiency than those of vertebrates [49,50]. In the insect CNS, acetylcholinesterase (AChE) breaks down the normally occurring transmitter acetylcholine, and this ends nerve signalling in the normal synaptic transmission between nerve cells [50]. However, neonicotinoids which bind to AChR cannot be broken down by AChE, and this leads to overstimulation of the sensory system of insects, paralysis and eventually death [49,51]. The selectivity of neonicotinoids for CNS of insects has been credited to their binding to nAChRs, in which the negatively charged nitro- or cyano-groups of neonicotinoids interact with a cationic subsite within insect nAChRs [52]. Due to the mechanism of action of neonicotinoids and conserved neurophysiology during the evolution of other animal phyla, some impacts of neonicotinoids on other non-target invertebrates, including molluscs, are expected [53], although at varying degrees among species [54]. Structural-binding analysis has demonstrated the binding of neonicotinoids to acetylcholine binding proteins from two molluscs species, *Lymnaea stagnalis* and *Aplysia californica* [55].

**Table 1 toxics-09-00021-t001:** Summary of neonicotinoid insecticides showing their properties, release year and some common trade names.

Neonicotinoids Properties	Imidacloprid	Acetamiprid	Nitenpyram	Thiamethoxam	Thiacloprid	Clothianidin
Released year	1991	1995	1995	1998	2000	2002
Molecular formula	C_9_H_10_ClN_5_O_2_	C_10_H_11_ClN_4_	C_11_H_15_ClN_4_O_2_	C_8_H_10_ClN_5_O_3_S	C_10_H_9_ClN_4_S	C_6_H_8_ClN_5_O_2_S
Molecular weight (g/mol)	255.7	222.67	270.72	291.71	252.72	249.68
Vapour Pressure (mm Hg)	1 × 10^−07^	4.5 × 10^−05^	8.2 × 10^−12^	6.6 × 10^−06^	6 × 10^−12^	1.3 × 10^−07^
Hydrolysis half-life at pH 7 (days)	>2000	na	na	≥572	10 to 63	na
Octanol-water coefficient (K_ow_)	3.7	6.27	−0.66	−0.13	1.26	5
Henry’s constant (atm m^3^/mole)	6.5 × 10^−11^	7.92 × 10^−08^	7.9 × 10^−11^	4.7 × 10^−10^	1.08 × 10^−14^	2.9 × 10^−16^
Melting point (°C)	136.4 to 143.8	98.9	~82.8	139.1	136	176.8
Anaerobic aquatic half-life (days)	27.1	45	~3	35.5	>365	27
Aqueous photolysis half-life (hours)	1 to 4	>34	~4.4	≥3.36	42	<24
Water solubility (mg/L at 20 °C)	510 to 610	4200	5.7 × 10^+05^	4100	185	327
Soil photolysis half-life (days)	38.9	25.1	1 to 15	47 to 54	na	34
Field dissipation half-life (days)	26.5–229	<18	<4	72 to 111	19	2 to 27
Soil adsorption coefficient (K_d_)	0.956–4.18	<4.1	na	0.59 to 2.03	na	0.62 to 1.94
Trade names	Confidor Merit Gaucho Admire Kohinor Prothor Advantage Gaucho Spectrum Premise Winner	Assail Intruder Adjust Rescate Tristar Saurus Prize Tristar Mosiplan Gazelle Trivor	Capstar Bestguard	Actara Cruiser Helix Platinum Centric Maxide Meridian Flashship Endigo Optigard Durivo Agri-flex Voliam	Calypso Biscaya Viper Piranha	Aloft Clutch Arena Votico Prosper Belay

na = not available. Data sources: [16,28,41,56,57].

### 2.2. Environmental Fate

Over the last decade, concerns with respect to the fate and impacts of neonicotinoids include persistence in the soil, water solubility and the likelihood of contaminating untreated zones within treated seed sowing areas, posing risks associated with the frequent use of neonicotinoids [23]. Numerous studies have also shown that neonicotinoids are highly persistent compounds (see reviews [58,59]), and residues of neonicotinoid insecticides can be detected in agricultural fields for up to a year after applications (Table 1), with the likelihood of contaminating other environmental compartments, including adjacent lands [60] and aquatic ecosystems [23,57]. The persistence and the likelihood of off-site contamination could expose non-target organisms to effective concentrations of neonicotinoids.

Neonicotinoids vapour pressure, soil adsorption coefficient and Henry’s law constant are very low (Table 1), an indication of low volatility and low air-borne particle dispersal [61,62]. This means that neonicotinoids will, in all probability, only be present in the air for a brief period during spraying and these compounds are not usually detectable in the air following applications by any currently available monitoring technique [63]. In air monitoring studies across four different counties in California, air samples were collected before, during and immediately after foliar spray applications of a neonicotinoid (imidacloprid) for the management of glassy-winged sharpshooter, *Homalodisca coagulata*. The results confirmed that imidacloprid residues were not detected in air samples collected in Santa Clara County [64], Solano County [65], Imperial County [66], and Butte County [67]. However, neonicotinoids are capable of contaminating adjacent fields through drift movement, or other mechanical methods (see review [59]). For example, planting neonicotinoid-coated seeds with a machine can release residues of the insecticide contaminated particulate matter into the adjacent environment [68,69]. Greatti et al. [70] and Krupke et al. [71] analysed plant and soil samples close to a field that was sown with neonicotinoids-treated seed and found varying concentrations of imidacloprid, clothianidin, thiamethoxam and their metabolites. Similarly, Biocca et al. [72] reported varying concentration (225–247 ng/m^3^) of clothianidin drifted from the planting of clothianidin-treated seeds. These examples confirm the potential for neonicotinoids to contaminate the environment beyond the immediate treatment area and highlight the need to follow best agricultural practices [73].

The persistence of neonicotinoids in soil and water depends on several factors including organic carbon content of the soil, temperature, pH (Table 1), frequency and quantity of neonicotinoids used, microbial community composition and function, and exposure to sunlight (Table 1). The mobility of neonicotinoids is lower in soil with high organic matter, attributed through the binding of the functional groups of the insecticides to the carboxylic acidic and phenolic hydroxyl groups of the soil organic matter [74]. Furthermore, the breakdown of plant materials that have been treated with neonicotinoids can release residues back into the soil [75], and this could pose risks to detritivores, for instance, terrestrial gastropods [76], in the environment.

The rate of degradation of neonicotinoids varies according to microbial community composition and function [77]. Using controlled experimental conditions in the laboratory, Anhalt et al. [78] reported that *Leifsonia* strain PC-21 degraded up to 58% from 25 mg/L imidacloprid within three weeks in trypsin solution containing 1 g/L succinate and D-glucose at 27 °C while the control (i.e., without *Leifsonia* strain PC-21) had no degradation of imidacloprid. Other microorganisms, for instance, *Pseudomonas* sp. [79], *Hymenobacter latericoloratus* [80], *Sphingomonas* sp. [81], *Bacillus aerophilus* and *Bacillus alkalinitrilicus* [82], have been reported to degrade neonicotinoids.

The rate of degradation of neonicotinoids also increases with sunlight and temperature [83]. For instance, in research conducted by Tisler et al. [84], there was no variation in the concentration of analytical grade imidacloprid in distilled water that was kept in the dark at fridge temperatures (3 ± 2 °C) over 22 days, but samples kept in room light at 21 ± 1 °C revealed decreasing concentrations over time. Furthermore, Lu et al. [85] reported that the rate of degradation of neonicotinoids (imidacloprid, acetamiprid, clothianidin, thiacloprid and thiamethoxam) in water was very fast in the presence of light approximating full-spectrum sunlight and negligible when the light source was attenuated [85], a situation that normally occurs in deeper waters and/or shallow waters with particulate matter obstructing light penetration. This suggests that (except for other degradation parameters) aquatic organisms inhabiting dark regions could be exposed to higher concentrations of neonicotinoids for a considerably longer time compared to organisms inhabiting the surface of aquatic systems.

## 3. A Literature Search of Neonicotinoids Studies on Molluscs

Using the Web of Science^TM^ and Scopus^®^ online databases, a detailed literature search of review articles and journal articles in English from 1995 to 2020 (25 years) that have been peer-reviewed was completed. The search terms included were pesticide, neonicotinoid, insecticide, mollusc, toxicity, impact, effect, lethal, sublethal, EC_50_, LC_50_, aquatic, terrestrial. This was structured as “Mollusc* OR mollusk AND neonicotinoid OR imidacloprid OR thiamethoxam OR clothianidin OR acetamiprid OR thiacloprid OR nitenpyram”. The search was repeated using the main classes, gastropod*, bivalv*, cephalopod*, scaphopod*, polyplacoph* and monoplacophor*, in place of mollusc*. Articles that were eligible for inclusion in this study were determined after a thorough review of their titles and abstracts. Papers were only included in this review if they reported experimental or field studies that documented the effects of neonicotinoids on at least one mollusc species or directly investigated impacts of neonicotinoids on some aspect of molluscan biology. Additionally, the reference lists in selected papers on neonicotinoids impacts on molluscs were searched, and the relevant papers were included in this article.

Using these terms, a total list of 38 (33 journals and five reviews) articles of molluscs that reported the impacts of either or both the active compound and a commercial formulation of neonicotinoids were reviewed. The results show an increasing trend in studies, with no study between 1995–2005 and the majority of the studies reported within the last five years (2015–2020) (Figure 2). While there are five review papers that mention neonicotinoids and molluscs, none to date have focused on molluscs and their responses to neonicotinoids. Previous reviews on the risks of neonicotinoids to a range of invertebrates, are dominated by studies on arthropods (e.g., [48,57]), but have concluded that molluscs are comparatively less sensitive to neonicotinoids.

## 4. Impacts of Neonicotinoids in Molluscs: What Do We Know

From the list of 33 journals articles recovered in our searches, 16 (48%) studies were on gastropods and 17 (52%) on bivalves (Figure 3A). In terms of habitat, five (17%) studies were on terrestrial molluscs and 24 (83%) in aquatic species; 10 (35%) in saltwater and 14 (48%) in freshwater (Figure 3B). Regarding the study type, 24 (77%) were lab-based investigations, two (7%) field studies and five (16%) mesocosm/microcosm experiments (Figure 3C). Grouping the studies based on specific neonicotinoids, there were 19 (54%) studies on imidacloprid (making it the most widely tested among the neonicotinoids), followed by thiamethoxam with 10 (29%) studies (Figure 3D) while six (17%) studies were in other neonicotinoids. In all the molluscs studied, 14 were on commercial species (i.e., mostly oysters and mussels) (Table 2). Of these commercially important species, only two investigated uptake and accumulation in the flesh of the exposed species, and these were in lab-based experiments (Table 2).

Most of the toxicity testing on the impacts of neonicotinoids on molluscs has focused on imidacloprid, and model species of molluscs (Table 2). The reason for the relatively high number of studies on imidacloprid toxicity to molluscs compared to other neonicotinoids could be because imidacloprid was the first neonicotinoids to be released (Table 1) and it is produced and used more than other neonicotinoids [86]. Some molluscs (oysters and mussels) are recognised as sentinels for ecotoxicological laboratory studies and allow for some comparison among other species of molluscs. However, this approach does not take into account other factors that may influence outcomes, particularly in field situations. Evolutionary lines of species, the sensitivity of species, exposure conditions and inter-species interactions are among the factors that could influence the results of toxicological studies. Risk assessment is usually based on ecotoxicological studies of at least three to five taxonomic groups, and species sensitivity distribution curves that are used to set guideline values are stronger when they include more taxonomic representations. If risk assessment relies on only a standard test species, we would misrepresent some sensitive species. For example, the arthropod *Daphnia magna*, the most commonly used aquatic animal representative for ecotoxicological tests is insensitive to neonicotinoids with 96 h LC_50_ of ≥44 mg/L see review—[28]. Therefore, there is a need to include molluscs in the species sensitivity distribution assessment process.

Historically, pesticide pollution studies were carried out by measuring concentrations in abiotic matrices and comparing these with tissue concentrations in animal models. Although this method is still useful for determining the concentration of pesticides in all matrices, it does not evaluate the toxic effects of chemicals on the exposed organisms [87]. As a result, monitoring the response of organisms using various biological markers, for instance, oxidative markers, is now being combined with the traditional method of contaminant monitoring [88]. Some of these oxidative stress markers, particularly antioxidant systems, are induced after acute exposure of bivalves [35,89] and gastropods [90] to neonicotinoids. However, following exposure to higher concentrations to neonicotinoids, or with a longer exposure time, the activities of antioxidant enzymes become inhibited in both mussels [32] and oysters [31]. Incorporation of these sublethal biomarkers into monitoring programs, along with traditional environmental concentration detection, could provide an indicator of potential environmental stress, that could subsequently lead to adverse health effects on populations of molluscs and other species if the stressor is not mitigated. This type of monitoring could be applied in areas used for mollusc aquaculture to provide an early warning system, so long as baseline data of healthy populations is available for comparison. However, follow up manipulative experiments in the laboratory will still be required to distinguish cause and effect, as well as for establishing effective doses that induce oxidative stress under controlled environmental parameters.

**Table 2 toxics-09-00021-t002:** Summary of previous journal articles that investigated neonicotinoids impact on molluscs species. * = life stage not reported, na = not available.

Species (Life Stage)	Species Class (Habitat)	Location (Study Type)	Neonicotinoids	Dose (Duration of Exposure)	Findings	Reference
**Accumulation**						
*Saccostrea glomerata* (adult)	Bivalve (saltwater)	Australia (lab-based)	Imidacloprid and a formulation (spectrum 200SC)	0.01, 0.05, 0.1, 1 and 2 mg/L (2 weeks)	Accumulated in the gill (0.40 µg/g at 0.01 mg/L and 4.6 µg/g at 2 mg/L exposures), adductor muscle (0.41 µg/g at 0.01 mg/L and 7.14 µg/g at 2 mg/L exposures) and digestive gland (0.22 µg/g at 0.01 mg/L and 4.39 µg/g at 2 mg/L exposures)	Ewere et al. [31]
*Saccostrea glomerata* (adult)	Bivalve (saltwater)	Australia (lab-based)	Imidacloprid	0.2 mg/L (1 to 3 days)	Accumulated in the gill (0.7 µg/g), adductor muscle (1.2 µg/g) and digestive gland (0.4 µg/g).	Ewere et al. [30]
*Deroceras reticulatum* (*)	Gastropod (terrestrial)	USA (lab- and field-based)	Thiamethoxam and a formulation (CruiserMaxx^®^)	0.08 and 0.15 mg/seed (8 days)	Accumulated residues and up to 0.5 µg/g in field-collected samples.	Douglas et al. [29]
**Behaviour**						
*Helix aspersa* (adult)	Gastropod (terrestrial)	Algeria (lab-based)	Thiamethoxam	100 and 200 mg/L (1 week)	Significant inhibition of locomotion and feeding at either concentration	Hamlet et al. [91]
*Crassostrea gigas* (larvae)	Bivalve (saltwater)	France (lab-based)	Imidacloprid	20 µg/L (24 h)	No effect on locomotion	Kuchovská et al. [92]
*Deroceras reticulatum*, *Arion distinctus and Milax gagates* (adults)	Gastropod (terrestrial)	UK (lab- and field-based)	Imidacloprid formulation (Gaucho)	0.7, 1.4, 2.8 g/kg seed (4 to 11 days)	Significant reduction in feeding on winter wheat at ≥2.8 g/kg (lab-based) and ≥0.7 g/kg (field-based)	Simms et al. [93]
*Corbicular fluminea* (larvae)	Bivalve (freshwater)	China (lab-based)	Imidacloprid	0.02, 0.2 and 2 mg/L (30 days)	Inhibition of feeding at 2 mg/L and burrowing at 0.02 mg/L exposures	Shan et al. [89]
*Unio tumidus* (adult)	Bivalve (saltwater)	Poland (lab-based)	Thiacloprid	10 µg/L (168 h)	Reduction of shell opening level and increase of shell opening rate	Chmist et al. [94]
*Saccostrea glomerata* (adult)	Bivalve (saltwater)	Australia (lab-based)	Imidacloprid	0.125 to 2 mg/L (1 to 4 days)	Reduction of filtration rate at 2 mg/L (day 1) and 0.5 and 1 mg/L (day 4)	Ewere et al. [30]
**Physiology**						
Immunity						
*Mytilus galloprovincialis* (adult)	Bivalve (saltwater)	Italy (lab-based)	Imidacloprid formulation (Calypso 480 SC)	7.77 and 77.7 mg/L (96 h and 20 days)	Increased hemocytes mortality rate, and reduction in hemolymph Cl^−^ and Na^+^	Stara et al. [32]
*Crassostrea gigas* (larvae)	Bivalve (saltwater)	Australia (lab-based)	Imidacloprid	0.001 mg/L (53 h)	No increased susceptibility to disease caused by OsHV-1	Oliver et al. [95]
*Saccostrea glomerata* (adult)	Bivalve (saltwater)	Australia (lab-based)	Imidacloprid	0.01 to 1 mg/L (96 h)	Increase in hemocyte counts and decrease in hemocytes aggregation at ≥0.1 mg/L exposure, but no effect of phagocytosis and hemocytes different types	Ewere et al. [35]
Growth and morphology						
*Marisa cornuarietis* (embryo)	Gastropod (freshwater)	Germany (lab-based)	Imidacloprid	Up to 50 mg/L (9 days	No effect on the formation of eye and tentacles, hatching, as well as weight post-hatching	Sawasdee and Köhler [96]
*Crassostrea gigas* (larvae)	Bivalve (saltwater)	France (lab-based)	Imidacloprid	0.2–2000 µg/L (30 h)	Increased in percentage of abnormal larvae at ≥200 µg/L	Kuchovská et al. [92]
*Saccostrea glomerata* (adult)	Bivalve (saltwater)	Australia (lab-based)	Imidacloprid and a formulation (spectrum 200SC)	≤2 mg/L (2 weeks)	No significant effect on the condition index	Ewere et al. [31]
*Crassostrea virginica* (*)	Bivalve (saltwater)	Canada (flow-through)	Thiamethoxam	119 mg/L (96 h)	No significant inhibition of shell growth	Finnegan et al. [97]
*Planorbella pilsbryi* (juvenile) and *Lampsilis fasciola* (juvenile)	Gastropod and bivalve (freshwater)	Canada (lab-based)	Imidacloprid and thiamethoxam	0.001 to 1 mg/L (28 days)	Significant reduction of growth and biomass production at ≥21 µg/L (imidacloprid) and ≥24.8 µg/L (thiamethoxam)	Prosser et al. [98]
Histopathology						
*Helix aspersa* (adult)	Gastropod (terrestrial)	Algeria (lab-based)	Thiamethoxam	10, 20 and 40 mg/L (20 days)	Degeneration of digestive tubules and breakdown of basement membrane in the hepatopancreas in all concentrations tested	Hamlet et al. [99]
*Parreysia cylindrica*	Bivalve (freshwater)	Indian (lab-based)	Thiamethoxam	14 mg/L (24 h, 96 h and 7 days) and 2.8 mg/L (21 days)	hypertrophy and sloughing of the epithelium, epithelial necrosis, tubular hyperplasia and rupture of the epithelial layer ≥14 mg/L. After 21 days, epithelial cells separated from the basement membrane	Patil [100]
*Mytilus galloprovincialis* (adult)	Bivalve (saltwater)	Italy (lab-based)	Imidacloprid formulation (Calypso 480 SC)	7.77 and 77.7 mg/L (10 and 20 days)	Caused alteration in the gills and digestive gland at the concentrations tested	Stara et al. [32]
*Corbicular fluminea* (larvae)	Bivalve (freshwater)	China (lab-based)	Imidacloprid	0.02, 0.2 and 2 mg/L (30 days)	Gill and digestive tissue damage at ≥0.02 mg/L, with severe damage at ≥2 mg/L	Shan et al. [89]
*Helix aspersa* (adult)	Gastropod (terrestrial)	Algeria (lab-based)	Thiamethoxam	25–200 mg/L (6 weeks)	Increase in the number of excretory vacuoles, breakdown of basement membrane and degeneration of digestive cells of the hepatopancreas at ≥25 mg/L	Hamlet et al. [91]
**Biochemistry**						
Synaptic connection						
*Lymnaea stagnalis* (larvae)	Gastropod (freshwater)	The Netherlands (lab-based)	Imidacloprid	0.001, 0.01 and 0.1 mg/L (10 days)	Increase in choline and acetylcholine turnover at ≥0.01 mg/L	Tufi et al. [101]
*Lymnaea stagnalis*	Gastropod (freshwater)	Japan (lab-based	Imidacloprid	na	Significant low affinity to the nicotinic acetylcholine receptor, possibly due to the presence of two orientation on the receptor for imidacloprid binding	Tomizawa and Casida [102]
*Lymnaea stagnalis*	Gastropod (freshwater)	Hungary (lab-based)	Acetamiprid formulations (Mospilan), imidacloprid formulation (Kohinor), thiamethoxam (Actara) and thiacloprid (Calypso)	0.01 and 0.1 mg/mL (5 s)	Each of the neonicotinoids inhibited the VD4-RPeD1. Calypso block 90% of excitatory postsynaptic potentials	Vehovszky et al. [103]
Cholinesterase						
*Mytilus galloprovincialis (adult)*	Bivalve (saltwater)	Italy (lab-based)	Imidacloprid and thiacloprid	0.1, 1 and 10 mg/L (96 h)	Reduction of acetylcholinesterase activity in the gill at ≥0.1 mg/L imidacloprid exposure and thiacloprid caused induction of acetylcholinesterase activity at 1 mg/L exposure and inhibition at 0.1 and 10 mg/L exposure	Dondero et al. [104]
*Helix aspersa* (adult)	Gastropod (terrestrial)	Algeria (lab-based)	Thiamethoxam	25–200 mg/L (6 weeks)	Inhibition of acetylcholinesterase activity at ≥25 mg/L, with higher concentrations causing greater inhibition	Smina et al. [90]
*Saccostrea glomerata* (adult)	Bivalve (saltwater)	Australia (lab-based)	Imidacloprid and formulation	0.01, 0.05, 0.1, 1 and 2mg/L (2 weeks)	Reduction of acetylcholinesterase in the gill at 2 mg/L	Ewere et al. [31]
*Biomphalaria straminea* (adult)	Gastropod (freshwater)	Argentina (lab-based)	Acetamiprid and a formulation (Assail 70^®^ WP)	150 and 1500 µg/L (14 days)	No effect on cholinesterase activity	Cossi et al. [105]
*Saccostrea* sp. (adult)	Bivalve (saltwater)	Colombia (lab-based)	Imidacloprid formulation (Imidogen 350 SC)	0.1, 1, 10 and 100 mg/L (96 h)	Reduction in total cholinesterase and eserine-sensitive cholinesterase activity in gill at ≥10 mg/L, and increase in eserine-sensitive cholinesterase activity in the digestive gland at 100 mg/L. Reduction of eserine-resistance cholinesterase activity in the adductor muscle at 10 mg/L	Moncaleano-Niño et al. [106]
*Corbicular fluminea* (larvae)	Bivalve (freshwater)	China (lab-based)	Imidacloprid	0.2, 0.2 and 2 mg/L (30 days)	Reduction of acetylcholinesterase in the gill 2 mg/L and digestive gland at ≥0.2 mg/L	Shan et al. [89]
Antioxidant activity						
*Corbicular fluminea* (larvae)	Bivalve (freshwater)	China (lab-based)	Imidacloprid	0.02, 0.2 and 2 mg/L (30 days)	Increase in glutathione S-transferase, catalase and superoxide dismutase activities in the gill and digestive gland at ≥0.02 mg/L	Shan et al. [89]
*Saccostrea glomerata* (adult)	Bivalve (saltwater)	Australia (lab-based)	Imidacloprid	0.01, 0.1 and 1 mg/L (96 h)	Increase in hemolymph glutathione S-transferase activity at ≥0.1 mg/L	Ewere et al. [35]
*Helix aspersa* (adult)	Gastropod (terrestrial)	Algeria (lab-based)	Thiamethoxam	25–200 mg/L (6 weeks)	Induction of glutathione S-transferase and catalase activity at ≥25 mg/L, with higher concentrations causing greater induction	Smina et al. [90]
*Biomphalaria straminea* (adult)	Gastropod (freshwater)	Argentina (lab-based)	Acetamiprid and a formulation (Assail 70^®^ WP)	150 and 1500 µg/L (14 days)	Significant increase in carboxylase activity, glutathione S-transferase activity, glutathione content, and decrease/inhibition of catalase activity, oxygen species levels and superoxide dismutase at both concentrations	Cossi et al. [105]
*Mytilus galloprovincialis* (adult)	Bivalve (saltwater)	Italy (lab-based)	Imidacloprid formulation (Calypso 480 SC)	7.77 and 77.7 mg/L (20 days)	Significant reduction of superoxide dismutase in the digestive gland and reduction of catalase activity in the gills at both concentrations	Stara et al. [32]
*Saccostrea glomerata* (adult)	Bivalve (saltwater)	Australia (lab-based)	Imidacloprid and formulation (spectrum 200SC)	0.01, 0.05, 0.1, 1 and 2mg/L (2 weeks)	Reduction of catalase in the digestive gland at 0.1 mg/L and glutathione S-transferase activities in the gill and digestive gland at ≥0.01 mg/L	Ewere et al. [31]
Energy reserve						
*Helix aspersa* (*)	Gastropod (terrestrial)	Algeria (lab-based)	Thiamethoxam	25–200 mg/L (6 weeks)	Reduction of total lipid content at ≥100 mg/L	Hamlet et al. [91]
*Saccostrea glomerata* (adult)	Bivalve (saltwater)	Australia (lab-based)	Imidacloprid and formulation (spectrum 200SC)	0.01, 0.05, 0.1, 1 and 2mg/L (2 weeks)	Altered the ratios and major classes of fatty acids at ≥0.01 mg/L	Ewere et al. [31]
*Lymnaea stagnalis* (larvae)	Gastropod (freshwater)	The Netherlands (lab-based)	Imidacloprid	0.001, 0.01 and 0.1 mg/L (10 days)	Decrease of fatty acids, possibly due to a downregulation of fatty acids biosynthesis. Exposure also caused an upregulation of lipids at ≥0.01 mg/L	Tufi et al. [101]
*Helix aspersa* (*)	Gastropod (terrestrial)	Algeria (lab-based)	Thiamethoxam	25–200 mg/L (6 weeks)	Reduction of tissue carbohydrate and protein contents at ≥100 mg/L	Hamlet et al. [91]
Omics						
*Mytilus galloprovincialis (adult)*	Bivalve (saltwater)	Italy (lab-based)	Imidacloprid and thiacloprid	2 mg/L (96 h)	Upregulation of heat shock proteins gene, protein translation genes, and downregulation of chitinase, endo-beta-glucanase, scavenger receptor cysteine-rich partial and profoldin subunit 4	Dondero et al. [104]
*Corbicular fluminea* (larvae)	Bivalve (freshwater)	China (lab-based)	Imidacloprid	0.02, 0.2 and 2 mg/L (30 days)	Downregulation of multixenobiotic resistance and heat shock protein genes at ≥0.02 mg/L	Shan et al. [89]
*Saccostrea glomerata* (adult)	Bivalve (saltwater)	Australia (lab-based)	Imidacloprid	2mg/L (96 h)	Upregulation of cargo and scavenger receptor activity-related genes and downregulation genes involved in axoneme, cilium or flagellum-dependent cell motility, dephosphorylation and phosphatase activity	Ewere et al. [30]
*Crassostrea gigas* (larvae)	Bivalve (saltwater)	France (lab-based)	Imidacloprid	0.2–2000 µg/L (72 h)	Upregulation of SOD [Cu/Zn], genes coding for two metallothioneins (*mt1* and *mt2*), and downregulation of SOD [Mn], genes linked with apoptosis and cell cycle regulation at ≥10 µg/L	Kuchovská et al. [92]
*Saccostrea glomerata* (adult)	Bivalve (saltwater)	Australia (lab-based)	Imidacloprid	0.01, 0.1 and 1mg/L (96 h)	Expression of several hemolymph proteins, including the upregulation of severin, heat shock proteins, superoxide dismutase and calmodulin, and the downregulation of collagens, actins, myosin heavy chain and CEP209_CC5 domain-containing protein ≥0.01 mg/L	Ewere et al. [35]
**Population dynamics**						
*Marisa cornuarietis* (embryo)	Gastropod (freshwater)	Germany (lab-based)	Imidacloprid	50 mg/L (9 days)	No effect on mortality	Sawasdee and Köhler [96]
*Melanoides tuberculatus* (adult), *Melanoides tuberculatus* (juvenile), *Lamellidens marginalis** and *Viviparous bengalensis**	Bivalve and gastropod (freshwater)	Bangladesh (microcosm)	Imidacloprid	0.003–3 µg/L (2–23 days)	No effect on mortality/population	Sumon et al. [107]
*Lampsilis fasciola* and *Planorbella pilsbryi* (juvenile)	Bivalve and gastropod (freshwater)	Canada (lab-based)	Imidacloprid, thiamethoxam, clothianidin, acetamiprid and thiacloprid	0.01–10 mg/L (7 and 28 days)	No reduction in viability at low concentrations. The estimated 7 days LC50 for the first three neonicotinoids on the list was ≥4 mg/L, and the 28 days LC50 was ~182 µg/L	Prosser et al. [98]
*Physella acuta** and Sphaeriidae*	Bivalve and gastropod (freshwater)	Spain (mesocosm)	Imidacloprid and neonicotinoids mixtures (containing imidacloprid, acetamiprid, thiacloprid, clothianidin and thiamethoxam)	0.2–250 µg/L (0–56 days)	Significant increase in the number of Sphaeriidae 250 µg/L imidacloprid treatment and a decrease of *Physella acuta* 250 µg/L neonicotinoids mixtures	Rico et al. [108]
*Physa* sp.**, Lymnaea* sp.*, *Planorbis* sp.* and *Musculium lacustre**	Bivalve and gastropod (freshwater)	UK (mesocosm)	Thiamethoxam formulation (Actara^®^ 25 WG)	1–100 µg/L (0–92 days)	No effect on mortality/abundance	Finnegan et al. [109]
*Lampsilis siliquoidea* (juvenile and adult) and *Villosa iris* (glochidia)	Bivalve (freshwater)	Canada (lab-based)	Imidacloprid, clothianidin and thiamethoxam	0–21 mg/L (24 h for glochidia and 28 days for juvenile and adult))	Only 8% decrease in glochidia viability at the maximum concentration tested. Clothianidin exposure at >9 mg/L caused 22% mortality in juvenile *Lampsilis siliquoidea*	Salerno et al. [110]
*Biomphalaria straminea* (adult and juvenile)	Gastropod (freshwater)	Argentina (lab-based)	Acetamiprid and a formulation (Assail 70^®^ WP)	150 and 1500 µg/L (14 days for adult and 30 days for juvenile)	No effect on mortality	Cossi et al. [105]
*Lymnaea stagnalis* and *Radix peregra* (*)	Gastropod (freshwater)	Canada (lab-based)	Thiamethoxam	100 mg/L (48 h)	No effects on mortality and immobilisation	Finnegan et al. [97]
*Planorbella trivolvis* and *Physella acuta* (*)	Gastropod (freshwater)	USA (mesocosm)	Clothianidin formulation (Arena)	0.6, 5 and 352 µg/L (48 h)	No significant effect	Miles et al. [111]
*Deroceras reticulatum* (*)	Gastropod (terrestrial)	USA (field-based)	Thiamethoxam and a formulation (CruiserMaxx^®^)	≥0.152 mg/seed (2 weeks after seed emerge)	Increase in population density due to reduction in predation or predators density	Douglas et al. [29]
*Radix* sp. (*)	Gastropod (freshwater)	Germany (microcosm)	Imidacloprid	0.6–40 µg/L (7 weeks)	Increase in the population at 40 µg/L, probably due to a decrease in competition from other sensitive species	Colombo et al. [112]

### 4.1. Neonicotinoids Accumulation

Studies on neonicotinoid accumulation within molluscs species, and the possibility of transmission to higher consumers along the food chain, due to consumption of the exposed molluscs are inadequate (Table 2). The likelihood for a pesticide or other environmental toxicants to accumulate in molluscs is dependent on the chemical properties of the toxicant [113,114], concentration in the environment, exposure duration [115], the sensitivity of the species [116], metabolism in the tissues [117,118] and depuration kinetics [119]. Various factors, including uptake and accumulation in specific tissues or organs and surface adherence, facilitate the accumulation of environmental toxicants [120]. Neonicotinoids have low octanol-water partition coefficient (Table 2) and as such, are not predicted to cross lipid bilayers. However, research has shown uptake and cellular accumulation of neonicotinoids, possibly due to the interaction of neonicotinoids with other macromolecules in cells [121]. For example, Douglas et al. [29] reported accumulation of the neonicotinoid (thiamethoxam) in the tissue of field-collected and lab-exposed species of *Deroceras reticulatum*. Using lab-based approaches, Ewere et al. [30] and Ewere et al. [31] reported the accumulation of imidacloprid in different tissues of the commercially important bivalve (*Saccostrea glomerata*) at concentrations that were higher than the regulatory guideline for seafood (0.05 mg/kg) in Australia. This could have major implications for food quality and safety because neonicotinoids and their metabolites have been shown to cross the blood-brain barrier in birds [122] and mammals [123] with subsequent negative effects [122,124], including death in humans (see review [34]). This means that commercially important molluscs could be devalued if chemical residues of neonicotinoids from upstream agricultural run-off and accumulate in bivalve growing area. These results suggest that the exposed molluscs could be a risk to organisms that consume molluscs along the food chain, including humans. Therefore, the level of neonicotinoids should be monitored in the environment, particularly in waters where bivalves are produced commercially.

Bioaccumulation can also have ecological consequences, for example, using a lab-based approach, Douglas et al. [29] reported that after the accumulation of thiamethoxam in the tissue of the agricultural pest slug *D. reticulatum*, the concentrations transmitted to the predaceous beetle impaired or killed more than half of the population of this beetle. Furthermore, their results showed that field-collected samples had decreasing concentration along the food chain (with primary consumers having higher concentrations of neonicotinoids), but the concentrations detected in *D. reticulatum* was high enough to kill all the insects that feed on the exposed slugs [29]. Neonicotinoids could also potentially accumulate in the aquatic food webs. For instance, crayfish and crab feed on molluscs and neonicotinoids have been reported to significantly cause stress and mortality to these crustaceans [33,125].

While there no reported studies on the elimination time of neonicotinoids from the tissue of terrestrial molluscs, these chemicals appear to be eliminated rapidly in aquatic species [32]. This may be due to the high solubility of neonicotinoids (Table 1) and the open circulatory system of mollusc species. Ewere et al. [30] reported the metabolism of imidacloprid to hydroxyl-imidacloprid in the digestive system and gills and imidacloprid-olefin in the gills of oysters. Their study also reported the elimination of imidacloprid from the gills and adductor muscles of oysters after as little as four days in imidacloprid-free water [30]. Fast elimination of neonicotinoids from bivalves, and potentially other aquatic molluscs, could reduce the time the organism is internally impacted [32] after run-off events, as well as reducing the potential for food chain transfer. However, elimination of neonicotinoids in terrestrial molluscs could be slower, leading to a delayed recovery post-neonicotinoid exposure, as reported in the land snail *Theba pisana* after exposure to thiamethoxam [126]. Further studies are required to understand the uptake and elimination (either through metabolism and/or depuration) of neonicotinoids in tissues of various groups of molluscs, as well as the recovery of the exposed molluscs.

### 4.2. Impact on Behaviour

Impact of neonicotinoids on the behaviour of molluscs has been inadequately studied and is only described in five studies (two in gastropods and three in bivalves) (Table 2). One of the observed effects of neonicotinoids on the behaviour of molluscs is the inhibition of movement in mobile aquatic and terrestrial species, at environmentally relevant concentrations (Table 2). Inhibition of movement in molluscs is a response to a wide range of stressors, including herbicides (e.g., [127]), pesticides (e.g., [128]) and other environmental contaminants (e.g., [129,130]). In Algeria, Hamlet et al. [91] reported that a neonicotinoid (thiamethoxam) significantly inhibited movement in the adult land snail *Helix aspersa* at ≥100 and 200 mg/L; concentrations that were lower than the application rate in Algeria. Similarly, Shan et al. [89] demonstrated that imidacloprid caused a significant reduction in burrowing of the bivalve *Corbicular fluminea* at concentrations as low as 0.02 mg/L. *C. fluminea* naturally burrow to escape predators and resist environmental stressors; therefore, inhibition of burrowing behaviour in waters contaminated by neonicotinoids would leave these clams more vulnerable to mortality. Impacts on movement can be impacted in mollusc species after exposure to environmentally relevant concentrations of neonicotinoids implies direct effects on the nervous system, and potentially a suite of other impairments of tertiary function that have not yet been explored.

In some species of molluscs (e.g., squid), reduced movements not only increases their exposure to predators but also reduces the ability to obtain food. Inhibition or reduction of feeding activity in molluscs could result from exposure to neonicotinoids. Hamlet et al. [91] reported that after exposure to 100 mg/L thiamethoxam the land snail *H. aspersa* did not feed, due to the inability of the gastropod to move to a close-by food source. Additionally, studying the destruction of winter wheat by slugs, Simms et al. [93] reported that environmentally relevant concentrations of a formulation of imidacloprid (Gaucho) caused a reduction in the feeding rate of *Deroceras reticulatum*, *Arion distinctus* and *Milax gagates*. In bivalve molluscs, imidacloprid has been reported to inhibit the feeding rate of *Corbicular fluminea* [89] and *Saccostrea glomerata* [30]. Overall, the limited data on the behavioural response of molluscs to neonicotinoids (Table 2), as well as small numbers of species of molluscs studied, suggests further research examining more behavioural responses across a wide range of molluscs behaviours is essential for understanding the extent of toxicity of neonicotinoids to molluscs.

### 4.3. Impacts on Physiology

Alteration to the physiology of molluscs has been acknowledged as a useful means for evaluating sublethal impacts from pesticides exposure [131], including neonicotinoids (Table 2). Neonicotinoids have been demonstrated to inhibit the cholinergic excitatory component of the VD4-RPeD1 connection in the CNS of a pond snail (*L. stagnalis*) [103], which could potentially lead to other physiological impacts. The growth of arthropods has been reported to be impaired by acute neonicotinoids exposure at a concentration as low as 1 µg/L (see review [132]). However, environmentally relevant concentrations of neonicotinoids do not appear to acutely impact the growth and morphology of mollusc species (Table 2). Exposure of up to 50,000 µg/L imidacloprid for nine days did not affect the weight, formation of eyes and tentacle in the embryo of a ramshorn snail, *Marisa cornuarietis* [96]. However, chronic or higher concentrations of neonicotinoids exposure significantly impact the growth and morphology of some molluscs. For example, 28 days exposure to imidacloprid (21 µg/L) and thiamethoxam (25 µg/L) caused a significant reduction in growth and biomass of juveniles *Planorbella pilsbryi* and *Lampsilis fasciola* [98]. Because of the few studies on the growth and morphological response of molluscs to neonicotinoids, more studies are required in order to ascertain whether longer exposure durations would affect the growth of other classes of molluscs.

The immune system is another sublethal indicator in molluscs that is very sensitive to stressors (e.g., [133,134,135]), including pesticides [35,136]. Only three studies have examined the effects of neonicotinoids on the immune system response of molluscs (Table 2), and those both assessed bivalves. Acute exposure to ≥0.1 mg/L imidacloprid caused an increase in hemocyte count and decreased hemocytes aggregation, but hemocyte phagocytosis was unaffected, in oyster *S. glomerata* [35]. Similarly, imidacloprid increased the mortality rate of hemocytes and caused a reduction of hemolymph Cl^−^ and Na^+^ in mussels *M. galloprovincialis* [32] suggesting an impact on osmoregulation. No increased susceptibility to oyster herpes virus (OsHV-1) was found in the larvae of oyster *Crassostrea gigas* after exposure to a very low concentration (1 µg/L) of imidacloprid for 53 h [95]. This suggests that some immune parameters of molluscs can be affected by neonicotinoid exposure, but because of the very limited data, additional studies are required to establish if neonicotinoids impact immune parameters and cause a reduction in the defence system of molluscs, leading to increased vulnerability to other environmental stressors.

Histological studies help to establish the causal relationships between the various biological processes in organisms and elevated contaminant exposure. They are sensitive tools to directly link the effect of a pollutant to direct impacts on organisms because exposure may induce lesions in the exposed tissue or organs of animals [137]. Both lab- and field-based approaches have been utilised to test the histopathological effects of neonicotinoids on mollusc species (Table 2). Hamlet et al. [91] and Hamlet et al. [99] reported an increase in the excretory vacuole, breakdown of the basement membrane, degeneration of digestive cells/tubules in the hepatopancreas of the land snail *H. aspersa* after exposure to thiamethoxam at ≥10 mg/L. While this exposure concentration seems high, it was within the application rate for this neonicotinoid in the country (Algeria). For aquatic species of molluscs, gill and digestive tissue damage were found in the larvae of *C. fluminea* after exposure to imidacloprid at concentrations as low as 0.02 mg/L [89], which is within the range of concentrations found in aquatic systems around the world (e.g., [11,28,57]). This means that exposure to low concentrations of neonicotinoids can damage the tissues of molluscs.

Physiological response of molluscs to neonicotinoids appear to depend on several factors, including the life stage of molluscs, the concentration of neonicotinoids, exposure pathway, duration of exposure and species of mollusc (Table 2). Therefore, more work will be needed to understand the physiological impact of neonicotinoids across a broad range of molluscs under various environmental conditions.

### 4.4. Impact on Biochemistry

Owing to the mode of action of neonicotinoid insecticides, the biochemical response of molluscs to neonicotinoids has been studied more than other endpoints (Table 2). There is substantial data on cholinesterase activity as a biochemical indicator for the effects of neonicotinoids in molluscs (Table 2), with significant inhibition of this enzyme depending on several factors, including the tissue, the concentration of neonicotinoids and duration of exposure (Table 2). While many mollusc species could potentially be exposed to effective concentrations of neonicotinoids in aquatic and terrestrial ecosystems, information on the cholinesterase response to neonicotinoids has been studied in only four bivalves and two gastropods molluscs (Table 2). Significant inhibition of acetylcholinesterase at ≥0.1 mg/L imidacloprid exposure was reported in bivalves hemolymph [35], gill and digestive gland [30,31,89,104], whereas the neonicotinoid (thiamethoxam) only inhibited this enzyme in the gastropod *H. aspersa* at ≥25 mg/L [90]. The inhibition of cholinesterase is similar to the effects of neonicotinoids reported in other invertebrates (see review [138]), implying that these insecticides are not specific to the CNS of target insects, as previously suggested [139]. Because the inhibition of this enzyme in molluscs can cause rapid spasms in voluntary muscles leading to subsequent biochemical, physiological and behavioural alteration and risk to predation [140], a wide range of biochemical endpoints should be investigated to assess the full magnitude of the response to chronic exposure low concentrations of neonicotinoids.

The oxidative stress response to neonicotinoids has been described in some molluscs, including the activities of catalase (CAT), glutathione (GSH), glutathione peroxidase (GPx), glutathione S-transferase (GST) and superoxide dismutase (SOD) (Table 2). The metabolic and physiological status of molluscs after exposure to environmental stressors can be assessed using these oxidative stress biomarkers (see review [141]). Exposure to low concentrations (≥0.02 mg/L) of imidacloprid caused an increase in GST activity in the hemolymph of *S. glomerata* after 96 h [35] and an increase in CAT, GST and SOD activities in gills and digestive gland of *C. fluminea* [89]. During xenobiotic metabolism in molluscs, there is an induction of these enzymes, which help to maintain homeostasis in the exposed cells [142,143]. This is because xenobiotic metabolism generates reactive oxygen species that can cause stress and apoptosis of cells [144]. The antioxidant enzymes have been described as detoxifying ‘batteries’ in invertebrates that have evolved to provide some protection [145]. However, prolonged or high concentrations of neonicotinoids caused the inhibition of these enzymes in the gill and digestive gland of two bivalves (*S. glomerata* [31] and *M. galloprovincialis* [32]). This suggests that neonicotinoids can overwhelm the oxidative stress systems and the protection offered by these enzymes in molluscs, depending on the concentration and the duration of exposure.

Significant inhibition of antioxidant enzymes could lead to a build-up of free radicals, which can ‘steal’ electrons from unsaturated fatty acids. This causes lipid peroxidation that can cause serious damage to the cell membrane [146]. When there is a significant depletion of lipids, catabolism of carbohydrates and proteins provides an energy source for the maintenance of metabolic needs [147]. Although there are no reported studies on the relationships between neonicotinoids exposure and lipid peroxidation leading to depletion of energy reserves, exposure to imidacloprid and thiamethoxam caused a reduction of the lipid, carbohydrates and proteins contents of *H. aspersa* [91] and altered the ratios and major classes of fatty acids in tissues of *S. glomerata* [31] and *L. stagnalis* [101]. Therefore, more studies are required to understand the relationships between neonicotinoids and antioxidant inhibition, especially those that lead to the depletion of energy reserved in molluscs.

More recently, in bivalve toxicology, omics approaches have been used to gain a more profound understanding of the mechanisms related to the physiological responses of molluscs to environmental disturbances [104,148]. Transcriptomic and proteomic responses of molluscs to neonicotinoids have been reported in just three studies (Table 2). Imidacloprid and thiacloprid caused the upregulation of several proteins, including heat shock proteins and several antioxidant proteins in *S. glomerata* [35] and *M. galloprovincialis* [104]. Additionally, imidacloprid caused the downregulation of multixenobiotic resistance and heat shock protein genes in *C. fluminea* [89] and several genes, including those involved in the axoneme, cilium or flagellum-dependent cell motility in *S. glomerata* [30]. These potential effects on cilia and flagella are of potential concern for digestion, sperm motility and larval motility and feeding. Therefore, omics approaches should be combined with traditional methods in order to have a deeper understanding of the physiological responses of molluscs to neonicotinoids, especially vulnerable early life stages and in tissues where pollutants accumulate or are metabolised.

### 4.5. Impacts on Population Dynamics

Neonicotinoids impacts on the population dynamics of molluscs have been studied in a broader suite of ecologically important species compared to other endpoints, but do not encompass any commercially important bivalve species. Field or microcosm studies have even shown a significant increase in the population of *Deroceras reticulatum* after treatment with thiamethoxam [29] and an increase in the population of *Radix* sp. after treatment with imidacloprid [112]. These two studies concluded that the increase in the population of these molluscs after neonicotinoids treatments was due to a significant reduction in competition and predation from other neonicotinoids sensitive species [29,112]. The majority of studies investigating the effects of neonicotinoids on population dynamics focus on early life stages (Table 2). This could be related to perceptions that embryos, larvae and juveniles are likely to be more vulnerable to field concentrations, whereas mortality in adult molluscs is not expected in comparison to other sensitive invertebrates (e.g., most arthropods) (see review [132]). Impacts of neonicotinoids on molluscs have shown that environmentally relevant concentrations of imidacloprid, clothianidin, thiacloprid and thiamethoxam under laboratory conditions do not induce mortality in the embryos of *Marisa cornuarietis* [96], or juveniles of *Lampsilis fasciola* and *Planorbella pilsbryi* [98,110]. While these studies have shown no negative effects at some life stages of molluscs, none address the full lifecycle and important metamorphic stages of molluscs that could be impacted by neonicotinoids. For instance, some molluscs (e.g., bivalves and many gastropods) undergo external fertilisation in which the gametes are released into the water column. There are currently no studies examining the impacts of neonicotinoids on mollusc gametes, such as sperm viability, membrane permeability, mitochondria polarisation and acrosomal membrane integrity or on fertilisation or metamorphosis or settlement success. These parameters in molluscs gametes are very sensitive to environmental stressors [149,150]. Therefore, furthers studies examining potentially more sensitive life stages of molluscs to neonicotinoids are required.

## 5. Synthesis and What We Still Need to Know

The neuroactive insecticide group known as neonicotinoids are used to control damaging insects in crops and improve food production. The increasing production and utilisation of these compounds have led to the contamination of aquatic and terrestrial ecosystems, thereby threatening productivity and survival of many non-target species, including molluscs. Some data exists on the impacts of neonicotinoids on molluscs (Table 2, Figure 4). A synthesis of biomarker responses of different species of molluscs to neonicotinoids provides some idea of the range of impacts that neonicotinoids can have on molluscs (Table 2). However, due to the species-specific responses, it is not possible to predict a comprehensive understanding of neonicotinoids toxicity to any particular mollusc species (Table 2). Nevertheless, understanding of the breadth of neonicotinoids impacts on molluscs can be achieved by comprehensive studies of certain species. For instance, Ewere et al. [30], Ewere et al. [31] and Ewere et al. [35] demonstrated tissue absorption and distribution of neonicotinoids in Syndey rock oysters, as well as a wide range of impacts on the behavioural, biochemical and physiological parameters (Figure 4). This indicates the complexity of effects that occur across different tissues in adults oysters exposed to imidacloprid. However, these studies did not evaluate neonicotinoid impacts on the early life stages of the oysters. Therefore, because of the gaps in knowledge (Table 3), the full threat that neonicotinoids pose to molluscs remains unclear. For this reason, more work is needed to identify and describe molluscs responses to various concentrations of neonicotinoids under different conditions and life stages (Table 3).

## 6. Conclusions

This review describes the current state of knowledge on the impacts of neonicotinoids on molluscs. The possible exposure routes of neonicotinoid insecticides to molluscs were identified including contaminated soil, food sources and water (for aquatic species). Acute and chronic exposure to effective concentrations of neonicotinoids is likely to occur for both aquatic and terrestrial molluscs, and evidence of accumulation in the tissue of mollusc was confirmed in a few studies. This exposure could significantly affect the behaviour, physiology, reproduction and productivity of mollusc species and data available indicate a cause for concern. Additionally, for edible species, there is the potential for human exposure through the consumption of neonicotinoids-exposed molluscs, and thus further studies are required to investigate the risk in commercial species that are grown in agricultural catchments. Rapid and reliable biomarkers include oxidative stress enzymes that can establish the sublethal effects of neonicotinoids in a range of molluscs. However, more in-depth studies across all scales, from genes to whole organism effects, encompassing all stage of the life cycle are still required to ascertain the full risk of environmental contamination by neonicotinoids to a broader range of commercial and ecologically important molluscs.

## Figures and Tables

**Figure 1 toxics-09-00021-f001:**
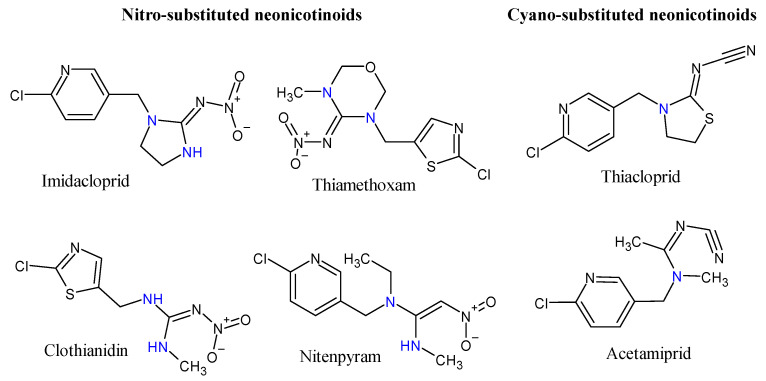
Structure of commercially available nitro- and cyano-neonicotinoids showing the amine groups (blue).

**Figure 2 toxics-09-00021-f002:**
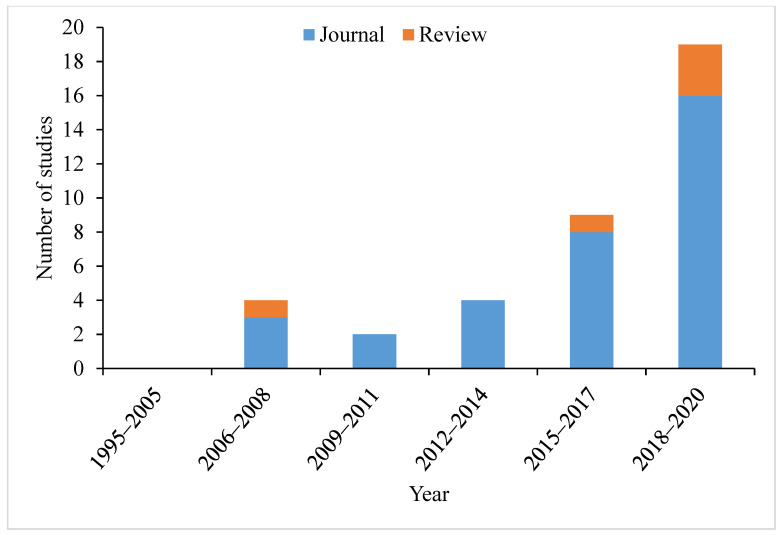
The number of review and journal articles on the impacts of neonicotinoids on molluscs showing increasing trend from 1995 to 2020.

**Figure 3 toxics-09-00021-f003:**
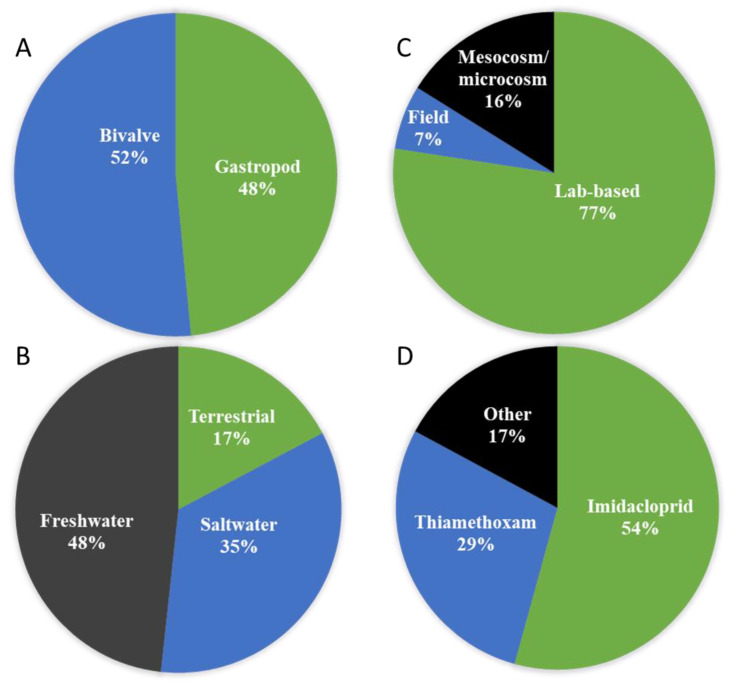
Summary of studies on the impacts of neonicotinoids on molluscs based on several factors: habitat type (**A**), study design (**B**), class of molluscs (**C**), and neonicotinoid type (**D**).

**Figure 4 toxics-09-00021-f004:**
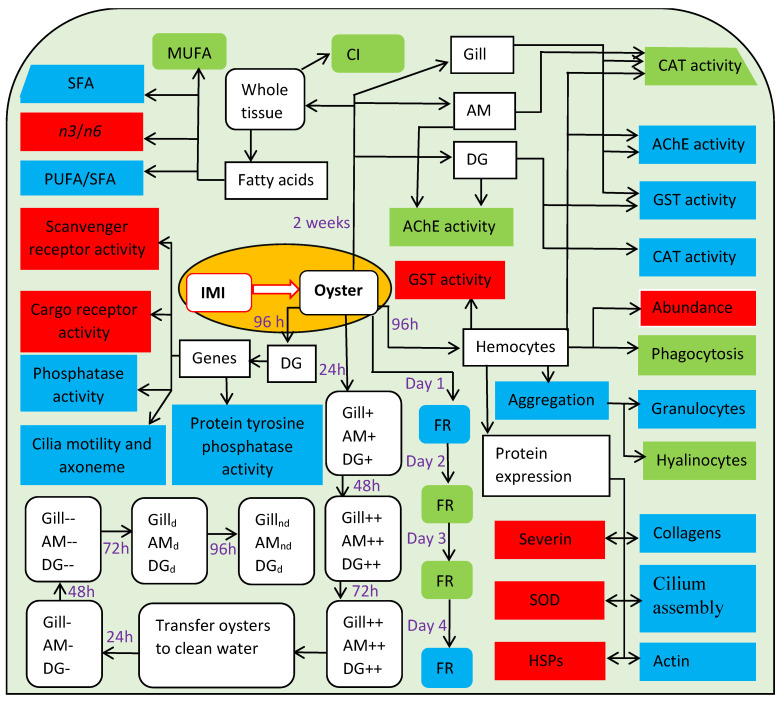
A conceptual network of the different parameters tested in a bivalve (oyster) after exposure to imidacloprid (IMI): Green = No effect; Red = Increase/elevation; Blue = Inhibition/Reduction; +, Absorbed; ++, Accumulated; -, Reduced concentration; --, Further reduction; d, Barely detected; nd, Not detected. SFA, Saturated fatty acid; PUFA, Polyunsaturated fatty acid; MUFA, Monounsaturated fatty acid; *n*-3, Omega 3 fatty acids; *n*-6, Omega 6 fatty acids; FR, Filtration rate; DG, Digestive gland; CI, Condition index; AChE, Acetylcholinesterase; CAT, Catalase; GST, Glutathione S-transferase; AM, Adductor muscle; HSPs, heat shock proteins; SOD, Extracellular superoxide dismutase. Source: Ewere [151].

**Table 3 toxics-09-00021-t003:** What we need to know about the impacts of neonicotinoids to molluscs and the possible approaches to address them.

Knowledge Gap	Further Research
No data on the impact of neonicotinoids on gametes or fertilisation success of molluscs.	Acute studies to identify the impacts of neonicotinoids to gametes and fertilisation success of molluscs, as well as acute and chronic trials on the development processes and metamorphosis of early life stages. For species that undergo external fertilisation (e.g., bivalves), there is also the need for studies determining the toxicity of neonicotinoids to sperms under natural environmental conditions.
Minimal data on the impacts of neonicotinoids on the safety and nutritional quality of edible molluscs.	Studies determining the acute and chronic effects of neonicotinoids on the safety and nutritional quality of edible species of molluscs, especially commercial species of gastropods and bivalves.
No data on the impacts of neonicotinoids on some ecologically important classes of molluscs.	Studies to determine the lethal and sublethal impacts of neonicotinoids to all the classes of molluscs, including classes that are not economically important.
Limited data on the impact of neonicotinoids mixtures on molluscs	Studies to identify the possible synergistic or antagonistic impacts of neonicotinoid mixtures as well as in combination with other chemicals and/or other environmental stressors on molluscs.
Limited data on the accumulation and elimination potential of neonicotinoids in molluscs.	More exposure experiments to determine the rate of accumulation, metabolisms and depuration of neonicotinoids in molluscs, to understand the risk of the possible exposure of other organisms higher along the food chain.
Very limited data exist on the impacts of neonicotinoids under stressful environmental regimes in molluscs.	Studies determining the effects of neonicotinoids on molluscs under various conditions, including salinity, temperature and pH.
Very limited data on the impacts of acute and chronic neonicotinoids exposure in molluscs under natural conditions	Mesocosms and field experiments to determine the impacts of neonicotinoids exposure to molluscs under natural conditions.
Very limited data exists on the genetic changes and regulatory mechanisms underlying molluscs response to neonicotinoids	Transcriptomics, DNA methylation and targeted gene expression studies to assess the physiological response of molluscs to neonicotinoids.
Limited data exist on the impacts of neonicotinoids on the physiology and immune system of molluscs.	Controlled manipulative studies to establish the causal effects on physiological and immunological responses of molluscs to neonicotinoids and any consequent tertiary effects on disease resistance, growth and mortality.
No data on the possible carry-over effects to the offspring due to adult exposure to neonicotinoids	Manipulative experiments to determine transgenerational impacts or resistance in offspring of molluscs that have been exposed to neonicotinoids.

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
