# Peer review of "Impacts of Neonicotinoids on Molluscs: What We Know and What We Need to Know"

_toxics, 2021, doi:10.3390/toxics9020021_

Round 1
Reviewer 1 Report
The authors in the manuscript entitled "Impacts of neonicotinoids on molluscs: What we know and what we need to know" carried out a literature review of the effects of different neonicotinoid concentrations on molluscs, including behavioral, physiological and biochemical responses. In addition, the authors in the review point to areas that require research in this area to provide a more complete understanding of the effects of exposure of molluscs to neonicotinoids. The manuscript submitted for evaluation is well structured in accordance with the review requirements. The authors correctly described the methodology used and based their study on 148 references. I believe that the study systematizes knowledge in this area and is very useful. I have only minor comments that the authors could consider:
- Are data available on the health risk assessment of the consumption of molluscs with residual neonicotinoids?
- Table 2 and for example line: 70, 336, 334, 367, 397 report the effect of different concentrations of neonicotinoids on molluscs. On the other hand, what are the concentration ranges for the use of neonicotinoids in crops - what are the maximum / recommended concentrations for a single use in the preparation of spray liquid solutions? Can these concentrations be compared?
- Are pesticides from the neonicotinoids group more dangerous to mollusc compared to other pesticide groups? Do the studies carried out so far suggest that the problem is solved, e.g. the choice of lower-risk pesticides - or maybe the use of biopesticides?
- Please correct minor errors in the bibliography to make it comply with the publisher's recommendations.
Reviewer 2 Report
Ewere et al. wrote a manuscript comprehensively reviewing the current status of the research on the effects of neonicotinoids on mollusks. I would like to make some comments on a few points that should be amended.
Major
Section 4.4: Physiological effects of neonicotinoids are not sufficiently described. It is desirable to discuss where neonicotinoids act in the nervous system of mollusks. I suppose that the neuromuscular junction is not cholinergic in mollusk, so neonicotinoids may exert their effects on somewhere else in the nervous system.
Minor
line 146: Is “contaminant” “contaminate”?
Figure 2: 1996-2005 may be 1995-2005.
Figure 3: The order of the panels (A, B, C, D) should be consistent with the appearance in the text above.
line 467: Some word seems to be lacking in the last of this sentence. physiological what? physiological traits? physiological aspects?
Reviewer 3 Report
toxics-1035159
Comments and Suggestions for Authors (toxics-1035159)
The article entitled “Impacts of neonicotinoids on molluscs: What we know and what we need to know” addresses a very interesting concerning residuals toxics from agricultural activities which are continually affecting the biodiversity worldwide. This article constitutes a warning to the authorities since traces or low concentration of neonicotinoids in the terrestrial and aquatic ecosystem can threat molluscs, but also other species of mollusc-eating arthropods. In general, the manuscript is well written, the introduction tackles the most important point about the history of neonicotinoids, distribution in the environment, and toxic effects upon target species and non-target species. These non-target species can include crabs, crayfish, bivalves, and humans. Among these species, the authors highlighted the putative effect of this pesticide on filter-feeding bivalves’ species. (e.g., oysters, scallops, mussels and clams) which have a very important ecological role in the shellfish reefs, being economically relevant species, as well. This background led the reader to the aim which is to review the response of molluscs to neonicotinoids. This approach could be helpful to find the main gaps in the knowledge of the toxicological effects of neonicotinoids, that can be used in future managements of this pesticide to protect non-target species.
Throughout the review, some sections, were properly chosen and organized surveying the main structural and mechanism of action, selectivity of the neonicotinoids and their environmental fate as well. The authors also explained in the details the methods/consideration used as a filtering criterion of literature reviewed. Then, a proper revision and discussion were carried out, covering reaching the aim of the article. Finally, the author gave the main finding and conclusions.
Minor comments:
Recommendations:
Keywords: the authors can use non-target species as keyword instead of Mollusc; Neonicotinoidsince these two words have already been included in the abstract.
Format, space before Figures: Please check typos like space, and give the spacing properly between Figures and text.
Line 230-231: Please, check the format of this sentence.
